# Epidemiology of Mortality Due to Prostate Cancer in Poland, 2000–2015

**DOI:** 10.3390/ijerph16162881

**Published:** 2019-08-12

**Authors:** Małgorzata Pikala, Monika Burzyńska, Irena Maniecka-Bryła

**Affiliations:** Department of Epidemiology and Biostatistics, the Chair of Social and Preventive Medicine of the Medical University of Lodz, 90-136 Lodz, Poland

**Keywords:** prostate cancer, neoplasms, years of life lost, mortality, epidemiology, trends, socioeconomic factors, Poland

## Abstract

The aim of the study was to assess trends in mortality and years of life lost due to prostate cancer (PCa) in Poland in 2000–2015. The crude death rates (CDR), standardised death rates (SDR), standard expected years of life lost per living person (SEYLL_p_) and per death (SEYLL_d_) values were calculated. Joinpoint models were used to analyse time trends. In the study period, 61,928 men died of PCa. The values of mortality rates in 2000 (per 100,000) were: CDR = 16.97, SDR = 16.17, SEYLL_p_ = 332.1. In 2015, the values of all rates increased: CDR = 26.22, SDR = 16.69, SEYLL_p_ = 429.5. However, the SEYLL_d_ value decreased from 15.62 to one man who died due to PCa in 2000 to 13.78 in 2015. The highest SEYLL_p_ values occurred in the group of men with primary education (619.5 in 2000 and 700.7 in 2015). They were respectively 2.24 and 2.96 times higher than in men with higher education (275.7 and 237.1). SEYLL_p_ values increased in urban areas (from 295.7 to 449.4), slightly changed in the rural areas (from 391.5 to 400.2). Unfavorable trends in mortality due to PCa in Poland require explanation of the causes and implementation of appropriate actions aimed at mortality reducing.

## 1. Introduction

Prostate cancer (PCa) is the second most common cancer and the fifth most common cause of death due to cancer in men worldwide. In 2018, approximately 1.3 million new cases of prostate cancer were detected and they contributed to 359,000 deaths [1].

It was observed that the incidence of PCa is highly diversified in terms of geographical regions but PCa-related mortality appears to be less diversified. PCa incidence rates varied more than 25-fold worldwide in 2012. The highest rates were observed in Australia/New Zealand (age-standardised rate 111.6 per 100 000), and the lowest in South-Central Asia (4.5). According to the most recent 10-year temporal data, most countries observed increased incidence of PCa. Asian, Northern and Western European countries noted the sharpest increase [2].

Poland is characterised with high dynamics of incidence which has particularly accelerated within the last 15 years. In 2000 in Poland, PCa was diagnosed in 5049 males (crude rate 26.9 cases, age-standardised rate 18.7 per 100,000 males), whereas in 2015, it was detected in 14,211 males (crude rate 76.4 cases, age-standardised rate 43.8 per 100,000 males) [3]. In the years 2000–2015, the annual percentage change (APC) was 7.1% (95% CI 6.7 to 7.5).

Similar growing trends regarding the incidence of PCa were observed in the 1980s in the US [4]; since the early 1990s, such trends have also been observed in five Nordic countries [5], Australia and Canada [6,7]. The increase in the incidence rates in these countries was associated with implementation of the prostate-specific antigen (PSA) in the blood serum as a screening test for early detection and diagnosis of PCa.

Previous research carried out in Poland shows that there are large and constantly increasing disproportions in mortality due to all causes of death among educational groups. Rate ratio (RR) showing the relation of standardised death rates (SDR) in the group of men with primary education to SDR in the group of men with higher education was 4.0. In 2011 its value has increased to 4.8 [8].

Poland belongs to a group of countries in which there have been marked differences in health status of urban and rural populations with disadvantage for rural ones for many years [9]. However, the results of the study carried out in 2002–2011 showed that urbanization differences are slowly disappearing. RR in the group of men living in the rural with regard to males living in the urban areas was 1.05 in 2002, and 1.09 in 2011 [10]. 

The aim of this study is to analyse changes in male mortality in Poland due to PCa in the period of 2000–2015, with special consideration given to differences regarding the education level and place of residence (urban–rural). In the study, apart from crude death rates (CDR) and standardised death rates (SDR), the standard expected years of life lost (SEYLL) index was also used. Measures expressing premature mortality in units of lost years of life are more and more frequently used to assess the health status of the population in international epidemiological literature [11,12,13,14,15,16].

## 2. Material and Methods 

The research project was granted an approval of the Bioethics Committee of the Medical University of Lodz on 22 May 2012 No. RNN/422/12/KB. The study material was a database including 5,996,489 death certificates of all inhabitants of Poland who died in the period of 2000–2015. Of this number, 61,928 men died of prostate cancer (according to the International Statistical Classification of Diseases and Health Related Problems—Tenth Revision—ICD-10 coded as C61). The data were provided by the Department of Information of the Polish Central Statistical Office. The procedure of coding causes of death in Poland is performed in a similar to the one carried out in the majority of countries in the world, based on the so-called primary cause of death, or the disease which triggered a pathological process, leading to death. 

Data on the number of Poland population in particular educational groups and depending on the place of residence come from the Survey of the Economic Activity of the Population based on representative sample of households. This study has been conducted since 1992 by the Central Statistical Office in accordance with the methodology of the International Labour Organization and consists of a nationwide household sample survey.

The authors calculated crude deaths rates (CDR) and standardised death rates (SDR).
$$CDR=\frac{k}{p}\times 100,000$$
where *k* is the number of prostate cancer deaths and *p* is the number of men 

The standardisation procedure was performed with the use of direct method, in compliance with the World Standard Population [17].
$$SDR=\frac{\sum_{i=1}^{N}\frac{ki}{pi}wi}{\sum_{i=1}^{N}wi}$$
where *k_i_* is the number of prostate cancer deaths in this *i*-age group, *p_i_* is population size of this *i*-age group, *w_i_* is the weight assigned to this *i*-age group, resulting from the distribution of the standard population, and *N* is the number of the age groups

Years of life lost were calculated and analysed by the method described by Christopher Murray and Alan Lopez in Global Burden of Disease (GBD) [18]. The SEYLL index (Standard Expected Years of Life Lost) is used to calculate the number of years of life lost by the studied population in comparison to the years lost by the referential (standard) population.

There are some methods of calculating lost years of life and the main difference between them is a point of reference (i.e., the level of mortality which is considered “ideal”). For European comparisons, the authors of “Health statistics—Atlas on mortality in the European Union” suggested adopting as a standard life tables of 15 old European Union member states [19]. However, the use of this method made it impossible to compare with non-EU countries that used a different standard [13,14]. On a global scale, by 2010, the expected years of life in Japan was adopted as the standard life span because it was the highest. In the GBD 2010 study, WHO experts recommend using life tables, based on the lowest noted death rate for each age group, in countries with population above 5 million [20].

In this study, the SEYLL index was calculated according to the following formula:$$SEYLL=\overset{l}{\underset{x=0}{\sum }}d_{x}e_{x}^{*}$$
where $e_{x}^{*}$ is life expectancy, based on GBD 2010 life tables, *d_x_* is number of deaths at age x, *x* is age at which the person died, and *l* is the last age which the population reaches

The authors also applied the SEYLL per person (SEYLL_p_) index, which is a ratio of SEYLL and the size of the population, calculated per 100,000 inhabitants, and the SEYLL per death (SEYLL_d_) index, being a ratio of SEYLL and the number of deaths due to a particular cause (i.e., it expresses the number of YLL calculated per one dead person) [21].

The analysis of time trends was carried out with joinpoint models and the Joinpoint Regression program, a statistical software package developed by the U.S. National Cancer Institute for the Surveillance, Epidemiology and End Results Program [22]. This method is an advanced version of linear regression, where time trend is expressed with a broken line, which is a sequence of segments joined in joinpoints. In these points, the change of the value is statistically significant (*p* < 0.05). The grid search method was chosen. Permutation tests were used to select the number of joinpoints. Fit of an uncorrelated errors model was used as autocorrelated errors option. The maximum number of joinpoints was 2. Minimum number of data points between two consecutive joinpoints was 4. We also calculated annual percentage change (APC) for each segment of broken lines and average annual percentage change (AAPC) for the whole study period with corresponding 95% confidence intervals (CI).

In order to compare the number of years of life lost by men due to prostate cancer in particular categories (i.e., education level or place of residence (city area/rural area)), the authors calculated rate ratio (RR), which is a ratio of SEYLL_p_ in less privileged groups and SEYLL_p_ in more privileged groups with corresponding 95% confidence intervals (CI).

## 3. Results

In 2000, in Poland, the number of PCa-related deaths was 3147; in 2015, it increased up to 4,876 (Table 1). In 2000, the crude death rate (CDR) was 16.97 per 100,000 males, whereas in 2015, the value was 26.22 (APC = 2.3%, *p* < 0.05) (Table 2). The standardised death rate (SDR) in 2000 was 16.17. Annual percentage change in 2000–2015 was negative and amounted to −0.5% (*p* < 0.05). Despite the downward trend, in 2015 the SDR value was higher than in 2000 and amounted to 16.69 per 100,000 males.

The increasing number of deaths due to PCs resulted in an increased number of lost years of life (SEYLL) in males in Poland. In 2000, the number of SEYLL was 49,159 years and in 2015, its value increased to 67,194 years. The SEYLL_p_ increased from 332.1 to 429.5 per 100,000 males. The average annual percentage change (AAPC) was 1.7% (*p* < 0.05). In 2000–2013, APC was 1.2%, while in 2013–2015, it increased to 5.2% (Table 2).

The largest groups (about 25% of all deaths due to PCa) were men aged 70–79 in 2000 and men over 80 years old in 2015 (Figure 1). These changes contributed to a gradual decrease in the number of lost years of life per one man who died of PCa. The value of SEYLL_d_ index dropped from 15.62 in 2000 to 13.78 in 2015 (APC = −0.6%, *p* < 0.05) (Table 1 and Table 2).

The authors observed huge differences in values of SEYLL_p_ indices and in the pace of their changes depending on the level of education. In the group of men with higher education, SEYLL_p_ in 2000 was 275.7 years; in 2015, its value decreased to 237.1 (Table 3). The APC for the period 2000–2015 was -1.6% (*p* < 0.05) (Figure 2). The highest SEYLL_p_ values in the entire analysed period were observed in men with primary education and they were the following: 619.5 in 2000 and 700.7 in 2015 (APC = 0.6%, *p* < 0.05). The gap regarding the number of lost years of life between men with primary and higher education also increased. The rate ratio (RR) was 2.24 in 2000 (*p* < 0.05) and 2.96 in 2015 (*p* < 0.05) (Table 3).

Very large changes, manifested by a rapid growth rate of SEYLL_p_ indices, were noted in the group of males with secondary education. In 2000, SEYLL_p_ in this group was 200.7, while in 2015, it was 394.4 (APC = 4.5%, *p* < 0.05). In 2000, SEYLL_p_ in the group of men with secondary education was lower than in the group of men with higher education (RR = 0.73, *p* < 0.05). Due to the growing trend in the group of men with secondary education and the declining trend in the group of men with higher education, RR in 2015 was 1.66 (*p* < 0.05).

Slightly smaller, but also statistically significant differences in PCa-related mortality, were observed for different places of residence. In 2000, SEYLL_p_ was higher in rural residents than in city dwellers (391.5 vs. 295.7, RR = 1.32, *p* < 0.05) (Table 3). A reverse trend was observed in the years 2000–2015, in both male groups. In the group of inhabitants of rural areas, SEYLL_p_ decreased annually at a rate of -0.2% (*p* > 0.05) (Figure 3). In males inhabiting urban areas, values of SEYLL_p_ indices were growing. The APC value in 2000–2012 was 2.0% (*p* < 0.05), while in 2012–2015, its value increased up to 6.8% (*p* < 0.05) (Figure 3). As a result of these changes, the SEYLL_p_ ratio in 2015 was 400.2 in rural areas and 449.4 in urban areas (RR = 0.89, *p* < 0.05).

## 4. Discussion

In Poland, in 2000–2015, the number of deaths due to prostate cancer increased. In 2000, 3147 males died due to this cause (6.6% of cancer deaths); in 2015, the number was 4,876 (8.8% of cancer deaths).

Prostate cancer mostly affects older age groups. The risk of incidence increases rapidly in the sixth decade of life and it becomes highest after the age of 75 [3]. The risk of death due to this cancer increases in the seventh decade of life. Males over 70 years constituted 5.9% of the population in 2000, while in 2015, the value was 7.6% of all men in Poland [23]. It can be therefore concluded that aging of the population of Poland is the main reason for the increase in PCa-related mortality. A comparison of statistically significant increase in the CDR rate (APC = 2.3%) with a statistically significant decrease in the SDR rate (APC = −0.5%) confirm this thesis.

Some epidemiological studies to the PCa risk factors include several environmental and dietary factors in addition to age that may affect the risk for developing this cancer. Among these factors, influence of obesity is the most documented [24]. Three meta-analyses indicate an increase in relative risk from 1.01 to 1.05 for each BMI increase by 1 kg/m^2^ [25,26,27]. Less significant are the results of studies linking the increased risk of PCa among people with high meat and dairy consumption [28], low physical activity [28,29] and smoking tobacco [30]. The results of the National Multicenter Health Survey WOBASZ II, conducted in Poland in 2013–2014, show that a lot of men lead an unhealthy lifestyle. Only 32.4% of them had normal weight, 43.2% were overweight, and 24.4% were obese [31]. Although dairy consumption was below normal, and in the group of men studied it was only 51% of the recommended value, meat consumption far exceeded the norm and amounted to as much as 173% of the recommended value [32]. Lack of any physical activity was declared by 37.2% of men in Poland, and 19.6% practiced physical activity occasionally [33]. 30% of men in Poland smoked tobacco [34].

Wong et al. classified Poland together with Brazil, the Czech Republic, France, Ireland, Israel, Italy, Japan, Netherlands, Spain, Switzerland and the UK to a group of countries with increasing incidence and decreasing mortality due to PCa [2]. However, mortality in Poland in this group was decreasing least rapidly. The most rapid decline was observed in Israel (AAPC = −3.9), France (AAPC= −3.9%) and the Czech Republic (AAPC = −3.8%). The most positive PCa trends were noted in three countries: Finland, Sweden and the US, where both morbidity and mortality are diminishing.

In Poland, as in other European countries, prostate cancer is a disease with a relatively good prognosis. A significant increase in survival rates has been observed. The 5-year relative survival rate for men with PCa diagnosed in 2003–2005 in Poland was 76.4% [35].

In comparison with rates for patients diagnosed in the years 2000–2002, the increase in the percentage of patients surviving 5 years was 11.2 percentage points and it was the highest for all cancers in the male group. It is difficult to clearly indicate reasons for such significant progress. Easy access to tests determining the PSA level in the blood serum, increasing awareness of patients as well as better awareness of this problem in primary care physicians make patients report to oncologists earlier. Significant improvement in 5-year survival has been observed. However, the EUROCARE 5 study reveals that this parameter is higher for Europe and for patients diagnosed in 2000–2007 it was 88.1% [36].

In our study, we applied years of life lost. Many authors claim in their publications that lost years of life are better measures than commonly applied mortality rates, because they better identify social and economic consequences of the phenomenon of premature mortality [11,12,13,14,15,16]. The absolute number SEYLL and the SEYLL_p_ index values increased in 2000–2015, which is associated with an increasing number of deaths due to PCa. On the other hand, the values of SEYLL_d_ indices per one man who died of PCa decreased, which is caused by a higher mean age of men with PCa and higher rates of 5-year survival. In 2000, the SEYLL_d_ index due to PCa was 15.62, whereas in 2015, it was 13.78. A comparison of 20 diseases, being the most common causes of death in Poland in 2011, reveals that with regards to the SEYLL_d_ value, PCa occupied only the 16th place in the male group [37].

Results of our research indicate that changes in SEYLL_p_ values differ significantly depending on the education level. In the group of men with primary and secondary education, we observe an increasing trend and the growth is more rapid in the group with secondary education. With regards to the group of men with higher education, SEYLL_p_ values are gradually decreasing. Educational differences in mortality trends have been well analysed in many studies. People with higher education generally enjoy a more stable job situation, higher income, better working and housing conditions. In addition, they are more aware of benefits of healthy lifestyle; they smoke less, more often follow diets and engage in various forms of physical activity [38,39,40,41,42]. What is disturbing, however, is the fact that disproportions between educational groups in Poland are gradually increasing [8]. Our own study confirmed that this also applies to mortality due to PCa.

Results of global research on differences in mortality due to malignant tumors, including PCa, between urban and rural residents are highly equivocal. Some studies showed higher mortality in rural residents who are made to cover long distances to get access to oncological services [43,44,45]. Results of other studies did not reveal a significant difference in mortality between males with prostate cancer inhabiting rural and urban areas [46,47,48,49]. According to some other studies, the mortality rate is higher in urban residents [50,51,52]. In Poland the differences in mortality due to all death causes between urban and rural residents are slowly disappearing, which is explained by increased access to new technologies and a similar lifestyle patterns in the rural and urban areas. However, in the case of malignant neoplasms, differences are marked with disadvantage for urban residents [9]. In our own study, we showed that this trend also applies to PCa. In 2000–2015, SEYLL_p_ values in rural areas were quite stable, while in the city they increased rapidly, particularly in 2012–2015. These differences can be explained primarily by changes in the age structure of rural and urban residents. The percentage of male rural residents above the age of 70 years changed slightly between 2000 and 2015 (6.6% in 2000 and 6.7% in 2015). However, the percentage of urban residents above the age of 70 years increased from 5.5% to 8.2%.

## 5. Strengths and Limitations

The strong point of the study is that the database used in the analysis contains information on all deaths of Polish residents in 2000–2016. Quality of the analyses performed on the mortality statistics depend on the completeness and accuracy of the information contained in the death certificate and the proper and precise description of the cause of death. Poland is a country with 100% completeness of death registration. In order to standardize death causes, which are subject to further statistical analyses, it was determined that the doctor who pronounces death is responsible for filling in the death card, into which he or she puts the primary, secondary and direct death cause, whereas qualified teams of doctors are responsible for coding death causes according to the ICD-10 classification.

The data relating to 2012 shows that the cause of more than 28% of deaths (about 109,000) were inaccurately described, however, in the majority of cases (78,500) it concerned deaths due to cardiovascular diseases [53]. Coding of malignant tumours does not raise any objections generally. There were some data gaps regarding education, however they concerned only 0.87% of death certificates. Data regarding the place of residence were complete in all study periods. Death cards with missing information about education were omitted from the analysis.

## 6. Conclusions

Despite decreasing values of standardised mortality rates due to PCa in 2000–2015, this type of cancer is still a major public health problem in Poland, mainly due to aging of the male population. The number of standard lost years of life is increasing in males with primary and secondary education as well as urban residents. High PCa-related mortality may be associated with late detection of this disease. Hence, there is a need to implement educational programmes, primarily aimed at the most vulnerable male groups.

## Figures and Tables

**Figure 1 ijerph-16-02881-f001:**
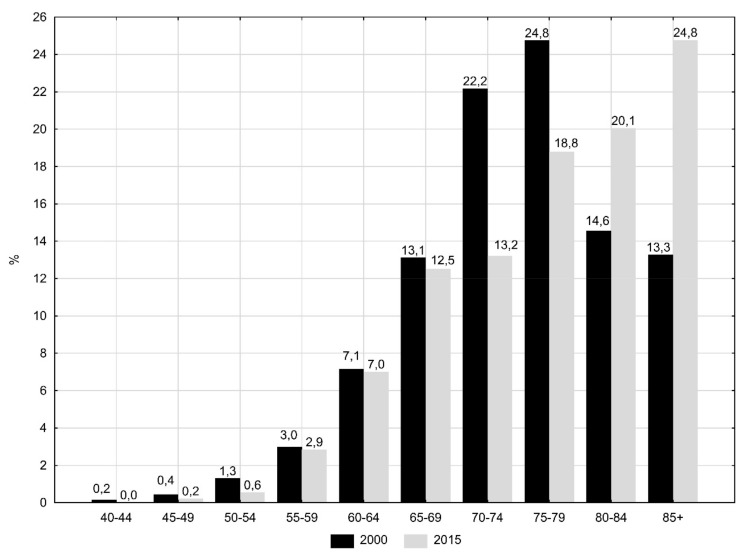
Percentage of deaths due to prostate cancer by age groups in 2000 and 2015 in Poland.

**Figure 2 ijerph-16-02881-f002:**
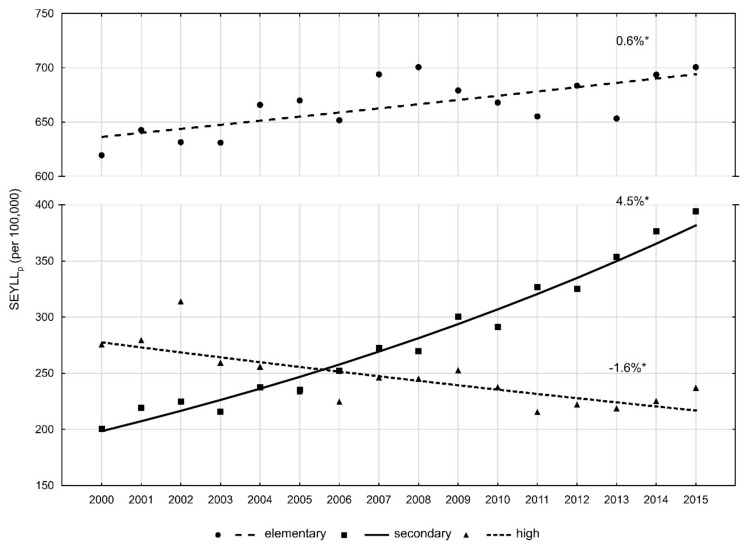
Time trends of the standard expected years of life lost per living person (SEYLL_p_) index due to prostate cancer by education in 2000–2015 in Poland.

**Figure 3 ijerph-16-02881-f003:**
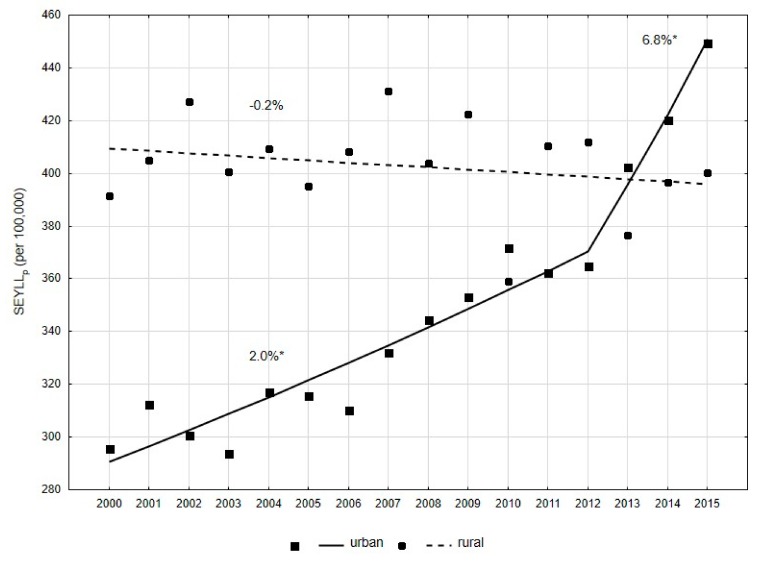
Time trends of the SEYLL_p_ index due to prostate cancer by place of residence in 2000–2015 in Poland.

**Table 1 ijerph-16-02881-t001:** Number of deaths and values of CDR, SDR, SEYLL, SEYLL_p_ and SEYLL_d_ due to prostate cancer in Poland in 2000–2015.

Year	Number of Deaths	CDR (Per 100,000)	SDR (Per 100,000)	SEYLL	SEYLL_p_ (Per 100,000)	SEYLL_d_ (Per Death)
2000	3147	16.97	16.17	49,156	332.1	15.62
2001	3365	18.15	17.04	51,893	347.8	15.42
2002	3488	18.85	17.23	52,425	349.0	15.03
2003	3390	18.34	16.41	50,621	334.9	14.93
2004	3578	19.37	16.85	53,621	352.7	14.99
2005	3592	19.46	16.29	52,944	346.4	14.74
2006	3681	19.98	16.35	53,416	348.2	14.51
2007	3932	21.36	16.87	57,037	370.7	14.51
2008	3892	21.14	16.19	56,727	367.8	14.58
2009	4042	21.93	16.40	58,841	380.6	14.56
2010	3940	21.12	15.50	57,423	366.9	14.57
2011	4085	21.90	15.55	59,776	381.5	14.63
2012	4199	22.52	15.55	60,136	383.6	14.32
2013	4281	22.98	15.49	61,413	392.0	14.35
2014	4440	23.85	15.61	64,314	410.7	14.49
2015	4876	26.22	16.69	67,194	429.5	13.78

CDR—crude deaths rates; SDR—standardised death rates; SEYLL—standard expected years of life lost; SEYLL_p_—standard expected years of life lost per living person; SEYLL_d_—standard expected years of life lost per death.

**Table 2 ijerph-16-02881-t002:** Time trends of CDR, SDR, SEYLL_p_ and SEYLL_d_ due to prostate cancer in Poland in 2000–2015 (joinpoint regression analysis).

Coefficients	Number of Joinpoints	Years	APC (95% CI)	AAPC (95% CI)
CDR	0	2000–2015	2.3 * (2.0; 2.7)	
SDR	0	2000–2015	−0.5 * (−0.8; −0.1)	
SEYLL_p_	1	2000–2013	1.2 * (0.8; 1.5)	1.7 * (0.8; 2.6)
		2013–2015	5.2 (−1.8; 12.6)	
SEYLL_d_	0	2000–2015	−0.6 * (−0.7; −0.4)	
SEYLL_p_ according to level of education				
high	0	2000–2015	−1.6 * (−2.4; −0.9)	
secondary	0	2000–2015	4.5 * (4.0; 4.9)	
elementary	0	2000–2015	0.6 * (0.2; 0.9)	
SEYLL_p_ according to place of residence				
urban	1	2000–2012	2.0 * (1.4; 2.6)	3.0 * (2.0; 4.0)
		2012–2015	6.8 * (1.5; 12.3)	
rural	0	2000–2015	−0.2 (−0.8; 0.3)	

* *p* < 0.05. CDR—crude deaths rates; SDR—standardised death rates; SEYLL—standard expected years of life lost; SEYLL_p_—standard expected years of life lost per living person; SEYLL_d_—standard expected years of life lost per death; APC—annual percentage change; AAPC—average annual percentage change; CI—confidence interval.

**Table 3 ijerph-16-02881-t003:** Standard expected years of life lost (SEYLL_p_) due to prostate cancer and rate ratio (RR) by level of education and place of residence, 2000 and 2015, Poland.

Risk Factors	Number of Deaths		SEYLL_p_		RR (95% CI)	
	2000	2015	2000	2015	2000	2015
Educational level						
high (ref)	217	566	275.7	237.1	1.0	1.0
secondary	1045	2474	200.7	394.4	0.73 * (0.70; 0.76)	1.66 * (1.62; 1.71)
elementary	1879	1717	619.5	700.7	2.24 * (2.17; 2.33)	2.96 * (2.88; 3.04)
Place of residence						
rural (ref)	1457	1802	391.5	400.2	1.32 * (1.30; 1.35)	0.89 * (0.88; 0.90)
urban	1690	3074	295.7	449.4	1.0	1.0

* *p* < 0.05. SEYLL_p_—standard expected years of life lost per living person; RR—rate ratio.

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
