# Peer review of "Epidemiology of Mortality Due to Prostate Cancer in Poland, 2000–2015"

_ijerph, 2019, doi:10.3390/ijerph16162881_

Round 1
Reviewer 1 Report
>> Small typographical errors need to be corrected, e.g. missing punctuation. Also language could be somewhat improved as some sentences are hard to follow.
>> The title of the article is a bit confusing. On the one hand you use the phrase “cross sectional” but on the other hand you state the range of years “2000-2015”.
>> “The PCa incident increased more than 25-fold worldwide in 2012”. In comparison to what?
>> “In 2000 in Poland, PCa was diagnosed in 5,049 males (26.9 cases per 100,000 males), whereas in 2015, it was detected in 14,211 males (76.4 cases per 100,000 males)”. Please specify if you give raw or standardized rates. If they are raw please also provide standardized rates.
>> “Years of life lost were calculated and analysed by the method described by Christopher Murray and Alan Lopez in Global Burden of Disease (GBD) [14]”. Publication (14) is not easily accessible to the international reader. Please add an appendix to the article in which you clearly describe formulas for this method.
>> “There are some methods of calculating lost years of life and the main difference between them is a point of reference”. Please be more specific here, give some examples.
>> “based on the lowest noted death rate for each age group, in countries with population above 5 million”. This seems like a very optimistic assumption. Please provide a short justification for this assumption. Please also discuss how sensitive are the results if you assume less optimistic scenario.
>> “In 2000, the highest percentage of PCa deaths, i.e. 47%, related to men aged 70-79. In 2015, the highest death rate, i.e. 44.9% related to men over 80 years of age (Figure 1).” This percentages seem not to agree with the graph.
>> Provide sizes for the subpopulations, e.g. how many rural, elementary persons were analyzed.
>> You may also want to show the results for education levels within urban/rural groups.
>> This seems to be a forced citation “In contrast, the value of this index for cervical cancer was respectively 28.10 and 23.70 years for one woman who died due to the above disease [21].”
Author Response
We are very grateful for insightful analysis of our work. We are also convinced that all the amendments will contribute to improve the quality of our manuscript.
Below are the answers for all of the Reviewer comments:
1. Small typographical errors need to be corrected, e.g. missing punctuation. Also language could be somewhat improved as some sentences are hard to follow.
A linguistic correction has been made.
2. The title of the article is a bit confusing. On the one hand you use the phrase “cross sectional” but on the other hand you state the range of years “2000-2015”.
We suggest changing the title to „Epidemiology of Mortality Due to Prostate Cancer in Poland, 2000–2015”
3. “The PCa incident increased more than 25-fold worldwide in 2012”. In comparison to what?
The difference of 25-fold concerned the comparison of the lowest and the highest incidence rates for prostate cancer worldwide in 2012. We agree that this sentence may have been insufficiently clear and we have changed it in the text into the following: “PCa incidence rates varied more than 25-fold worldwide in 2012. The highest rates were observed in Australia/New Zealand (age-standardized rate 111.6 per 100 000), and the lowest in South-Central Asia (4.5).”
4. “In 2000 in Poland, PCa was diagnosed in 5,049 males (26.9 cases per 100,000 males), whereas in 2015, it was detected in 14,211 males (76.4 cases per 100,000 males)”. Please specify if you give raw or standardized rates. If they are raw please also provide standardized rates.
The given values were for crude death rates. As suggested by the Reviewer, we explained this in the text and we added the values of age-standardized rates.
5. “Years of life lost were calculated and analysed by the method described by Christopher Murray and Alan Lopez in Global Burden of Disease (GBD) [18]”. Publication (18) is not easily accessible to the international reader. Please add an appendix to the article in which you clearly describe formulas for this method.
The literature position no. 18 has been changed to English reference.
6. “There are some methods of calculating lost years of life and the main difference between them is a point of reference”. Please be more specific here, give some examples.
As suggested by the Reviewer, the following part of the text describing two other methods of counting years of life lost was added: "For European comparisons, the authors of "Health statistics - Atlas on mortality in the European Union" suggested adopting as a standard life tables of 15 old European Union member states [19]. However, the use of this method made it impossible to compare with non-EU countries that used a different standard [13, 14]. On a global scale, by 2010, the expected years of life in Japan was adopted as the standard life span because just there it was the highest”.
7. “based on the lowest noted death rate for each age group, in countries with population above 5 million”. This seems like a very optimistic assumption. Please provide a short justification for this assumption. Please also discuss how sensitive are the results if you assume less optimistic scenario.
The adoption as the standard a life table based on the lowest observed death rate for each age group in countries of more than 5 million in population is a consequence of recommendations developed by the WHO team of experts working in the Global Burden of Disease 2010 study [20]. The construction of such theoretical life tables is, by nature, optimistic, because they are used to calculate lost years of life in relation to the theoretical population considered to be "ideal".
8. “In 2000, the highest percentage of PCa deaths, i.e. 47%, related to men aged 70-79. In 2015, the highest death rate, i.e. 44.9% related to men over 80 years of age (Figure 1).” This percentages seem not to agree with the graph.
The percentages given in the text concerned the combined age groups presented in the graph. Agreeing with the Reviewer that this could be a bit confusing for readers, we changed the sentence in the text into the following:
“The largest groups (about 25% of all deaths due to PCa) were men aged 70-79 in 2000 and men over 80 years old in 2015 (Figure 1)”.
9. Provide sizes for the subpopulations, e.g. how many rural, elementary persons were analyzed.
In the Table 3, a column with a number of males deaths per category for educational groups and places of residence was added.
10. You may also want to show the results for education levels within urban/rural groups.
The statistical data that we have regarding deaths due to PCs would allow us to carry out the analysis simultaneously for both variables (the level of education and place of residence). Data on the number of men in particular subgroups (needed as a denominator to calculate mortality rates) come from The Survey of the Economic Activity of the Population based on household population sample and do not include the number of men divided into the level of education and place of residence at the same time.
11. This seems to be a forced citation “In contrast, the value of this index for cervical cancer was respectively 28.10 and 23.70 years for one woman who died due to the above disease [21].”
This sentence has been removed from the text.
Reviewer 2 Report
In both the introduction and the discussion sections, the authors should focus more on the background in Poland, such as what has been known about how education and urban-rural residency affect the incidence and mortality in Poland, and what the current study would add to the existing literature; while reduce the comparisons in different parts of the world.
The Methods section lacked the background of the data base from which the study population was selected, the quality, and the period of the record, e.g. is it only available since 2000; how was the information on education level and urban-rural residence collected?
More details about the jointpoint regression should be added, such as the algorithms used, assumptions, and input parameters.
Since both the crude and adjusted death rates were calculated as mentioned in the Methods, the authors should be explicit about which rates were reported in the results (text and Figures).
Rate ratio comparing outcomes among education levels and between urban rural residences were reported in the results, but the modes used were not mentioned in the methods. Did the author adjusted for other important covariates such as age, and cancer presentation (state, histological type)?
The European Standard Population was used to calculate age-adjusted mortality rate, while the GBD2010 life tables were used to calculate the standard expected years of life lost. Would using the European population life table to calculate SEYYL be more consistent and preferable?
In the Table, a note should be given for the acronyms, or the author should spell out the full variable names used.
The discussion lacks a paragraph about the limitation of the study and about whether or not an Institutional Review Board approval of the study was needed or not, and whether an approval was obtained if needed.
Author Response
We are very grateful for insightful analysis of our work. We are also convinced that all the amendments will contribute to improve the quality of our manuscript.
Below are the answers for all of the Reviewer comments:
1. In both the introduction and the discussion sections, the authors should focus more on the background in Poland, such as what has been known about how education and urban-rural residency affect the incidence and mortality in Poland, and what the current study would add to the existing literature; while reduce the comparisons in different parts of the world.
The number of publications on differences in mortality between socio-economic groups in Poland is unfortunately not great. However, we have added a few paragraphs to the introduction and the discussion sections about Poland.
In Discussion section we have also added the part of the text discussing the prevalence of PCa risk factors in the population of Poland.
However, we would like to leave comparisons with other countries in the world, which in our opinion show the unfavorable trends in our country.
2. The Methods section lacked the background of the data base from which the study population was selected, the quality, and the period of the record, e.g. is it only available since 2000; how was the information on education level and urban-rural residence collected?
The period covered by the study is dictated by the number of analyzed records. The database of deaths of all Polish citizens in the years 2000-2016 contains over 6 million records. Obviously, the analysis period could be extended, but it is very difficult to process such a large amount of data. For this reason, we have decided to limit the analysis to the 21st century.
Regarding to information on the level of education and urban-rural residence, they come from The Survey of the Economic Activity of the Population based on representative household population sample.
This information has been added in the "Material and Methods" section.
3. More details about the joinpoint regression should be added, such as the algorithms used, assumptions, and input parameters.
In the "Material and methods" section the part of the text with detailed description of the joinpoint regression has been added.
4. Since both the crude and adjusted death rates were calculated as mentioned in the Methods, the authors should be explicit about which rates were reported in the results (text and Figures).
Both in the text and in the graphs descriptions, it is specified whether they concern crude or standardized mortality rates.
5. Rate ratio comparing outcomes among education levels and between urban rural residences were reported in the results, but the modes used were not mentioned in the methods. Did the author adjusted for other important covariates such as age, and cancer presentation (state, histological type)?
The use of death cards in statistical analysis has its limitations, but also advantages. The limitation is undoubtedly the number of information contained in the death card. The cause of death is coded using the International Statistical Classification of Diseases and Health Related Problem. Information on cancer presentation is therefore not available. However the unquestionable advantage of using death cards, is the completeness of the data, which we describe more extensively in the added section "Strengths and limitations".
6. The European Standard Population was used to calculate age-adjusted mortality rate, while the GBD2010 life tables were used to calculate the standard expected years of life lost. Would using the European population life table to calculate SEYLL be more consistent and preferable?
Adopting as a standard a life table based on the lowest observed death rate for each age group in countries of more than 5 million in population is a consequence of recommendations developed by the WHO team of experts working in the Global Burden of Disease 2010 study [20].
7. In the Table, a note should be given for the acronyms, or the author should spell out the full variable names used.
An explanation of acronyms has been added below each table.
8. The discussion lacks a paragraph about the limitation of the study and about whether or not an Institutional Review Board approval of the study was needed or not, and whether an approval was obtained if needed.
Section "Strengths and limitations" has been added. In the first paragraph of the "Material and methods" section, information on the consent of the Bioethics Committee of the Medical University of Lodz for conducting the study was also added.
Reviewer 3 Report
General comments:
In this manuscript, the authors investigated the trends of mortality and years of life lost due to prostate cancer using data from the death certificate. Although the rationale for the current paper is essential, however, the manuscript has some minor issues that need to be addressed before it gets to publish. I hope the comments will help the author to make it more clear.
Specific comments:
Abstract:
1) Results presented in the abstract is not too convincing. Please include data description in short and statistical analysis. Can the author present the result in more elaborately?
2) The conclusion looks not related to the findings. Please write it in line with the results.
Introduction:
3) The fifth paragraph is too long, consider it to shorten and include references in between a paragraph. The author stated, “Death rates for PCa have been decreasing in Northern America, Oceania, Northern and Western Europe, developed Asian countries and the United States, which has been linked to implementation of screening tests, which in turn, facilitated setting an earlier diagnosis and improved treatment.” This is a too long sentence. I suggest to break it down and include references for country-specific papers.
4) The last sentence of intro, "Many authors claim in their publications that lost years of life are better measures than commonly applied mortality rates, because they better identify social and economic consequences of the phenomenon of premature mortality”. This doesn’t fit here, consider to move or change somewhere else.
Materials and methods:
5) The author used death certificate data; however, it might be a better option to link death certificate data to hospital emergency data and hospital administrative data to look at PC mortality. This can also solve some unanswered questions. Please reconsider it to analyse or explain why you haven't done this.
6) To compare the number of years of life lost due to prostate cancer, why occupation has not been included in the model. I think socioeconomic variables are important too.
7) Although the nature of data looks longitudinal but author analysed in a cross-sectional way. Please reconsider to include time-varying factors in the mortality model. The author can also test mortality prediction model including factors responsible for PC deaths overtime.
Discussion:
8) Please explain in the discussion, why the number of deaths due to prostate cancer increases over time. What are the factors could responsible for these trends, except ageing?
9) Include strengths and limitations.
Author Response
We are very grateful for insightful analysis of our work. We are also convinced that all the amendments will contribute to improve the quality of our manuscript.
Below are the answers for all of the Reviewer comments:
Abstract:
1) Results presented in the abstract is not too convincing. Please include data description in short and statistical analysis. Can the author present the result in more elaborately?
The abstract has been changed according to Reviewer’s suggestions.
2) The conclusion looks not related to the findings. Please write it in line with the results.
Conclusions in the abstract have been changed as suggested.
Introduction:
3) The fifth paragraph is too long, consider it to shorten and include references in between a paragraph. The author stated, “Death rates for PCa have been decreasing in Northern America, Oceania, Northern and Western Europe, developed Asian countries and the United States, which has been linked to implementation of screening tests, which in turn, facilitated setting an earlier diagnosis and improved treatment.” This is a too long sentence. I suggest to break it down and include references for country-specific papers.
This paragraph has been removed according to suggestion of the other one Reviewer.
4) The last sentence of intro, "Many authors claim in their publications that lost years of life are better measures than commonly applied mortality rates, because they better identify social and economic consequences of the phenomenon of premature mortality”. This doesn’t fit here, consider to move or change somewhere else.
The last sentence of Introduction has been moved to Discussion.
Materials and methods:
5) The author used death certificate data; however, it might be a better option to link death certificate data to hospital emergency data and hospital administrative data to look at PC mortality. This can also solve some unanswered questions. Please reconsider it to analyse or explain why you haven't done this.
The data provided by the Polish Central Statistical Office is unfortunately not identifiable, which is caused by the personal data protection. Therefore, we can not link them to hospital databases in any way.
The use of death cards in statistical analysis has got undeniable advantages, mainly related to the completeness of data. We write about it more extensively in the Strengths and limitations subsection, added according to suggestion of the Reviewer.
6) To compare the number of years of life lost due to prostate cancer, why occupation has not been included in the model. I think socioeconomic variables are important too.
We fully agree with the Reviewer that the occupation is a very important socio-economic variable. Unfortunately, information about it is not included in the death cards in Poland. Until 2013, the source of income of died person’s was entered in the death cards, which was somehow connected with the occupation. Unfortunately the modified death cards do not contain this information.
However, the results of many scientific researches indicate that the most important indicator of socio-economic inequalities in population is the level of education. Other social characteristics, such as the occupation and amount of income, are largely explained by the level of education disparities. People with higher education generally have a more stable job situation, higher income, better working and housing conditions Therefore, education level may be treated as a synthetic variable.
7) Although the nature of data looks longitudinal but author analysed in a cross-sectional way. Please reconsider to include time-varying factors in the mortality model. The author can also test mortality prediction model including factors responsible for PC deaths overtime.
Conducted study is a longitudinal study, because its aim is to observe changes in mortality due to PCa in time. Cross-sectional analysis was also aimed at comparing different groups selected with the use of variables such as education level and place of residence at the same time points (mainly at the beginning and at the end of the analyzed period). In order to avoid misunderstandings, we have changed the title of the manuscript.
Our research is also a retrospective study. Of course, the designated trend lines might be used for forecasting in the short term, however this was not the aim of the study.
Discussion:
8) Please explain in the discussion, why the number of deaths due to prostate cancer increases over time. What are the factors could responsible for these trends, except ageing?
In the Discussion section, the part of the text describing the impact of PCa risk factors other than age and their prevalence in the population of Poland has been added.
9) Include strengths and limitations.
“Strengths and limitations” section has been added.
Round 2
Reviewer 1 Report
No further comments.
Author Response
Thank you very much for taking the time to read and review our manuscript.
Reviewer 2 Report
Were the comparison of SEYLLp among education levels and between urban/rural location conducted in the same model or separate models? Why choose only the 2000 and 2015 for comparison? Not 2000 through 2015? Are there any interactions between education level and urban/rural location? How were the differences in the age structure between urban and rural locations taken into account in the RR calculation?
European Standard Population was used to calculate age-adjusted mortality rate, while the GBD2010 life tables were used to calculate the standard expected years of life lost. Would using the European population life table to calculate SEYLL be more consistent and preferable? Or alternatively, using the GBD standard population to calculate age-adjusted mortality rate?
The discussions about time trend were mostly focused on comparing 2015 with 2000. If a different reference year was chosen, the results may be different. The annual percentage changes are more telling and should be presented in the abstract rather than the rates from the two endpoint years. For example, while “In 2015, the values of all rates increased: CDR=26.22, SDR=45.24, SEYLLp=429.5”, the APC for SDR was actually negative. Some discussion of the jointpoint regression results should also be incorporated. Another example, for SEYLLp, there appears to be two segments (2000-2013 and 2013-2015) for the overall SEYLLp, and two segments (2000-2012 and 2012-2015) for urban SEYLLp, suggesting that simply comparing 2000 and 2015 is not adequate.
Author Response
Were the comparison of SEYLLp among education levels and between urban/rural location conducted in the same model or separate models? Why choose only the 2000 and 2015 for comparison? Not 2000 through 2015?SEYLLp for the education level and urban/rural location were calculated for all years of the analyzed period. Their values are presented in figures 2 and 3. Joinpoint models are made for all years from 2000 to 2015 and present the annual percentage change in this period for each of the analyzed groups. The comparison between 2000 and 2015 (without presenting values for other years) was used only for RR coefficients. RR values change very slowly from year to year, that is why we decided to show only extreme years of analysis as more interesting.
Are there any interactions between education level and urban/rural location? How were the differences in the age structure between urban and rural locations taken into account in the RR calculation?In conducted analysis, we did not take into account the interaction between the level of education and urban/rural locations. We assumed that the analysis of mortality and related life years lost due to prostate cancer by the place of residence without taking into account other variables (including education) is important information for institutions responsible for conducting health policy and introducing educational programs. Education is a non-modifiable variable and territorial differences should be taken into account in order to carry out activities aimed at their gradual reduction. Thanks to such calculated indicators we inform what the current situation is, not about what it would be if the differences resulting from the education level didn’t exist. We hope the Reviewer will agree with our opinion.
There were no differences in the age structure between men in the urban and rural location. The structure similarity index was 94.3%.
European Standard Population was used to calculate age-adjusted mortality rate, while the GBD2010 life tables were used to calculate the standard expected years of life lost. Would using the European population life table to calculate SEYLL be more consistent and preferable? Or alternatively, using the GBD standard population to calculate age-adjusted mortality rate?As suggested by the Reviewer we have calculated the age-adjusted mortality rate using the World Standard Population.
The discussions about time trend were mostly focused on comparing 2015 with 2000. If a different reference year was chosen, the results may be different. The annual percentage changes are more telling and should be presented in the abstract rather than the rates from the two endpoint years. For example, while “In 2015, the values of all rates increased: CDR=26.22, SDR=45.24, SEYLLp=429.5”, the APC for SDR was actually negative. Some discussion of the jointpoint regression results should also be incorporated. Another example, for SEYLLp, there appears to be two segments (2000-2013 and 2013-2015) for the overall SEYLLp, and two segments (2000-2012 and 2012-2015) for urban SEYLLp, suggesting that simply comparing 2000 and 2015 is not adequate.The sentence quoted by the Reviewer comes from the Abstract, in which due to the limit of words we couldn’t explain in more detail the pace of change. However, in the calculations we took into account not only the extreme years 2000 and 2015. Detailed values for all the analyzed years are provided in the Table 1. In the text, every time we provide the value of the coefficients from 2000, annual percentage change and the final value from 2015. Where the trend has changed during the analyzed period, we emphasize this in the text, providing joinpoint and APC for each straight line broken:
“The SEYLLp increased from 332.1 to 429.5 per 100,000 males. The average annual percentage change (AAPC) was 1.7% (p <0.05). In 2000-2013, APC was 1.2%, while in the years 2013-2015, it increased to 5.2%”
“In males inhabiting urban areas, values of SEYLLp indices were growing. The APC value in 2000-2012 was 2.0% (p<0.05), while in 2012-2015, its value increased up to 6.8% (p<0.05) (Figure 3). As a result of these changes, the SEYLLp ratio in 2015 was 400.2 in rural areas and 449.4 in urban areas (RR = 0.89, p<0.05).”
To draw attention to the fact that SDR increased in 2015 compared to 2000 despite the decreasing APC rate we have added the following sentences in the Results section:
"The standardized death rate (SDR) in 2000 was 16.17. Annual percentage change in 2000-2015 was negative and amounted to -0.5% (p <0.05). Despite the downward trend, in 2015 the SDR value was higher than in 2000 and amounted to 16.69 per 100,000 males.”
Round 3
Reviewer 2 Report
The authors adequately addressed my concerns.